# Equity Market Description under High and Low Volatility Regimes Using Maximum Entropy Pairwise Distribution

**DOI:** 10.3390/e23101307

**Published:** 2021-10-05

**Authors:** Mauricio A. Valle, Jaime F. Lavín, Nicolás S. Magner

**Affiliations:** 1Facultad de Economía y Negocios, Universidad Finis Terrae, Santiago 7501015, Chile; 2Escuela de Negocios, Universidad Adolfo Ibáñez, Santiago 7941169, Chile; jaime.lavin@uai.cl; 3Facultad de Economía y Empresa, Universidad Diego Portales, Santiago 8370109, Chile; nicolas.magner@udp.cl

**Keywords:** return volatilities, maximum entropy principle, financial crisis, pairwise interactions, frustration, Kullback-Leibler divergence

## Abstract

The financial market is a complex system in which the assets influence each other, causing, among other factors, price interactions and co-movement of returns. Using the Maximum Entropy Principle approach, we analyze the interactions between a selected set of stock assets and equity indices under different high and low return volatility episodes at the 2008 Subprime Crisis and the 2020 COVID-19 outbreak. We carry out an inference process to identify the interactions, in which we implement the a pairwise Ising distribution model describing the first and second moments of the distribution of the discretized returns of each asset. Our results indicate that second-order interactions explain more than 80% of the entropy in the system during the Subprime Crisis and slightly higher than 50% during the COVID-19 outbreak independently of the period of high or low volatility analyzed. The evidence shows that during these periods, slight changes in the second-order interactions are enough to induce large changes in assets correlations but the proportion of positive and negative interactions remains virtually unchanged. Although some interactions change signs, the proportion of these changes are the same period to period, which keeps the system in a ferromagnetic state. These results are similar even when analyzing triadic structures in the signed network of couplings.

## 1. Introduction

The fluctuation of market asset prices are a good example of unpredictable time series and random processes, subject to complex interactions of a multitude of elements in a complex system under nonlinear dynamics [1]. Modeling equity market behavior without imposing assumptions on the dynamics and trading rules that govern it offers the opportunity to study this system as a set of stocks interacting with non-trivial rules. The Maximum Entropy Principle (MEP) allows modeling complex systems without prior assumptions as a naturalistic approach. This is done on the basis of incomplete information [2,3], taking a data-base approach to discover nonlinear interaction rules that provide information on the intrinsic nature of the whole system behavior [2]. This approximation results in a more simplified version of the actual underlying structure of the system that is consistent with the observed data. Unlike most of available literature studying interrelationships between financial assets, mostly based on linear correlations, our approach relies on a minimal probabilistic model with only pairwise interactions that accurately describe the financial system’s activity of asset return movements. The MEP method is not necessarily more appropriate than other methods to study the interrelationship between financial assets. However, it is an interesting complement to study them because it describes the behavior of returns and it suggests that the macroscopic behavior of the system is not necessarily microscopic but is governed by multiple nonlinear interactions.

The underlying assumption in the use of MEP is that the macroscopic behavior of a system is not microscopic in nature, but through the constellation of possible interactions and mutual influence of the units that make up the system [4,5,6]. In this line of thought, it would be possible to make an analogy of the financial system as a ferromagnetic Ising model [7], in which the state of each of the spins $s_{i}$ is subject to a local magnetic field, and subject to exchange interaction $J_{ik}$ with another neighboring spin *k* with state $s_{k}$. The Ising model makes explicit the fact that the behavior of an element is determined by how it is affected by other elements near it. Previous studies [2,8,9] show that this approximation correctly models the orientations and correlations of stock markets.

Under the MEP, it is necessary to describe the probability distribution of the system states from the observed data. Then, the model needs to be fitted by satisfying the first and second empirical moments of the distribution of states [8]. The advantage of this approach is that the fitted model satisfies the maximum entropy probability distribution [10]. Since entropy represents the lack of interaction between the spins, the MEP distribution is the least structured model possible.

New models have been developed that provide a more realistic approach to market behavior that outperform conventional methods by not relying on the premises of the efficient market hypothesis and the rational expectations of the agents. Some combine elements of Information Theory, Statistical mechanics, and particle interaction physics to explain the observed behavior of financial systems. Some of these models, such as the log periodic power law models of market crashes [11], heterogeneous agent models [12] and Quantal response statistical equilibrium model or QRSE [13], stand out for reproducing features that other conventional models do not capture, such as excess volatility and fat tails of distributions of returns. Among these models, the QRSE manages to capture and characterize the behavior of complex systems with a reduced set of parameters and proves to be the least biased as an inference method because it does not impose additional normative assumptions regarding the system components and lends itself very well to model the interaction between heterogeneous agents with limited rationality. For instance, in [14], the authors model the housing market dynamics at different stages of the housing market crash of 2018, capturing the characteristic patterns of boom-bust cycles without assuming in advance special features of the expectations of the special features of market agents’ expectations. They estimated an MEP and statistical equilibrium-based model. The distribution of system states contains all the information of the macroeconomic variables of the system being in equilibrium. Similary, [15] models the contagion propagation of the Italian interbank market using the MEP and data from the bilateral exposures for all Italian banks, again, without imposing assumptions regarding the behavior of the agents interacting in the system. Likewise, [16] studies the adoption of new technologies by cost-minimizing firms with limited capacity to process market information. In this case, the equilibrium distribution of the model estimated with MEP reproduces the observed pattern of technological change. Our study is not far from the QRSE-based models. However, unlike above studies, we recognize the connection with Ising’s physical-magnetic model, its equivalent with the state distribution of the system parameterized with couplings (equivalent to the equilibrium distribution), and the ferromagnetic behavior as a physical property of the system.

The financial market is a highly linked complex system with broad interconnections and interdependencies, where shocks easily turn into global events of systemic risk. Interconnectedness has a dual impact on systemic risk, for one hand, it could improve financial robustness when contributes to absorbs shocks, but for the other, it could generate contagion when propagates shocks [17]. At present, the interactions existent in financial markets are a relevant phenomenon in stock markets since contagion generates a significant change in stock’s correlation coefficients that turns into episodes of high synchronization of returns. This is a complex phenomenon with no single cause; on the contrary, as we observe during the initial stages of the COVID-19 outbreak its occurrence does not circumscribe to financial issues but is also related to multiple factors [18]. The synchronization of returns has multiples origins and implications. In a risk management context, this phenomenon is crucial. During high synchronization episodes, diversification losses its ability to suitably protect portfolios against losses, high synchronization periods tend to occur during market turmoil, precisely when investors most need the help of diversification as a tool to lessens the negative effects of shocks on their portfolios [19].

In this paper, we study the interaction structures of stocks present in periods that reveal important changes in the variability of asset returns. For this purpose, we build a model under the MEP in three different periods (before, during and after) for two major crises: the Subprime financial Crisis (SC) in year 2008–2009, and the financial crisis derived from the COVID-19 outbreak in year 2020 (CO). As we are dealing with interacting entities (indices and stock prices), we are interested in inferring the structure of interactions and external influences under financial shocks that bias the behavior of these entities’ movements in one or another direction. For the SC, we used market capitalization-adjusted daily indexes of the world’s largest stock market. For CO, we used the hourly prices of companies with the largest weight in the Dow Jones Industrial Average (DJIA), reflecting some of the largest companies in the US market. In this way, we can evaluate the MEP pairwise model to explain two systems with different data frequencies and in two different settings: in the first case, at the world market level, and in the second at the country level.

In this study, the MEP model is tested in two extraordinary events that put the financial system under stress and prompted a series of financial changes and monetary and fiscal interventions. It is important recognizing the different nature of the two shocks. The SC was a shock associated with banks runs and asset-prices crashes. The origin of this crisis is based on the lending of subprime loans (mortgage loans to individuals with no income or employment) from financial institutions and banks to individuals with low credit rating which are likely to default on a loan. In the SC, the bailouts were primarily focused on rescuing banks without capital to meet their financial obligations. On the other hand, the COVID-19 outbreak (CO) is a shock associated with infection rates, widespread lockdowns and spiking poverty [20] in which the financial sector is affected due to the impact of broad-based shutdowns (self-quarantine at home) and social-distances measures. In this case, these government responses produce an “induced coma” to the economy to reduce the spread of the virus. The effect is more sudden, radical and abrupt than in SC. At the financial level, the persistence of the COVID-19 outbreak keeps uncertainty in the economy and amplifies the volatility of the markets [21], the risks increase substantially, while the monetary authorities implement intensive policies to save the markets such as zero-zero interest rates and unlimited quantitative easing [22]. Unlike the SC, the CO involves unprecedented support not only to the financial system but also to households, firms, financial markets, and in general to the entire economic system affected by the lockdown. In CO the banks and the International Monetary Fund have provided macroeconomic stimulus in a variety of ways to help householders cope with job losses and lack of income, as well as small and medium-sized businesses to survive the lockdowns [23]. An essential difference between SC and CO is that the former is considered endogenous because its origin comes from inside, while the latter is exogenous because the origin of the problem is a pandemic and not in the economic-financial system. Nevertheless, both are extraordinarily volatile shocks that spured high levels of volatility in the financial markets.

Financial crises will continue to occur, and although they have been the subject of intense study, we believe that a closer look at the multitude of interactions that explain macroscopic behavior could offer greater insight into these rare events. Unlike other studies that focus on the emergence of collective behavior in financial crises and synchronization (for example, [5,24]), in this one we are interested in comparing the inter-relationship structure of the market and assessing how interactions change in two crises of a very different nature. In other words, we evaluate the dynamic nature of the interactions under shocks of different origin.

The many possible interactions between financial assets can be considered as a signed network in which interdependencies with positive and negative links co-exist. We consider this system’s characteristic, describing the sign and balance of the triads or triangles as system’s substructures. In physics, one aspect that identifies this class of disordered systems (with positive and negative interactions) is frustration, which plays a crucial role in the dynamics of complex systems [25].

Our main results indicate that the pairwise MEP distribution can successfully recover the average orientation of the spins and the correlation structure of the stocks and indices. However, we observe that the model’s ability to explain the information differs according to the crisis (SC or CO) we are studying. Interestingly, the magnitude change of interactions between these assets in periods of high and low volatility is minimal, and the level of frustration remains almost stable. This condition contrasts with large fluctuations in the orientation of spins and increased synchronization between spins in a high volatility period.

In Section 2 we describe the data and define the periods of high and low volatility for the Subprime and Pandemic crisis derived from the COVID-19 outbreak. We also evaluate the preservation of correlations and mean binarized returns. Finally, we describe the Boltzmann learning by which the parameter inference process is performed. In Section 3, we show the inference results and evaluate the consistency of the parameters to recover the moments of the returns. We also evaluate the ability of the pairwise model to explain the financial system and the the level of frustration of the system in each period with triadic configurations. Finally, in Section 4 we offer summary of results and discuss possible avenues of possible future research as well as practical implications in Section 5.

## 2. Materials and Methods

We have two different data sets: one for SC and the other for CO. For SC we study the dynamics of the values of 10 country market indices: four of Europe (UKX - United Kingdom, CAC - France, DAX - Germany and FTSEMIB - Milan), three of Asia (NKY - Japan, HSI - Honk-Kong and TWSE - Taiwan), two of Northamerica (SPX - SP500 of USA, SPTSX - Canada), and one of Southamerica (IBOV - Brazil). We analyze the daily closing values of each index from July 13, 2007 to January 15, 2010, which is equivalent to 656 days. For CO we study the price dynamics of 15 stocks representing 80% of the market capitalization of the Dow Jones. We analyze hourly prices from December 17, 2019 to April 6, 2020, which equals 788 prices over the period. The companies are: four of technology sector (Microsoft Corp., Apple Inc., Verizon Communications Inc. and Intel Corp.), three of retail sector (Home Depot Inc., salesforce.com Inc.; Walmart Inc.), two of banking sector (Visa Inc.; JPMorgan Chase & Co.), four of consumer goods (Johnson & Johnson, Procter & Gamble Corp., NIKE Inc.; Coca-Cola Co.), one of entertainment: Walt Disney Corp and one of health: United Health Group Inc. Table 1 shows a summary of the indices and stocks analyzed in this study.

Although there does not appear to be a formal definition of a financial crisis, in order to define the time segments that we will call a *crisis*, we will take an operational definition as a sudden and significant drop in market asset values over a long period of time. This definition is limited to a financial sense and does not take into account a broader sense that considers other factors of a macroeconomic nature. This broad definition allows us to compare the behaviour of the financial system in the crisis condition versus the condition just before and just after the prolonged and persistent collapse of asset prices. As a guide to identifying the periods, we use the S&P500, which is recognized as a good gauge of large-cap of U.S. equities and worldwide equity financial state.

The analysis periods cover both crises (SP and CO). We calculate the logarithmic returns of each index/stock *i* as $r_{i}\left(t\right)=\ln\left(v_{i}\left(t\right)\right)-\ln\left(v_{i}\left(t-1\right)\right)$ where $v_{i}\left(t\right)$ is the price/value of the index/stock *i* at time *t*.

Figure 1a shows the daily price evolution $v_{i}\left(t\right)$ of the S&P500 and other country indices (in gray) over an extended time span covering the SC. The same for the CO in $v_{i}\left(t\right)$ and other companies (in gray) in Figure 1b. In both cases there is a significant increase in the variability of returns $r_{i}\left(t\right)$ when the prices start to decrease drastically and persistently. Likewise, we observe a high level of volatility of returns for the period we can consider during the crisis. We calculate the estimated volatility of returns as $\sigma_{i}\left(t\right)={\left(\frac{1}{n-1}\sum_{t=1}^{n}\left(r_{i}\left(t\right)-\langle r_{i}\rangle \right)\right)}^{1/2}$ where *n* is the size of the roll window and $\langle r_{i}\rangle$ is the average of returns over that span of time. The volatilities $\sigma_{i}\left(t\right)$ on Figure 1 are computed using n=22 days for S&P500.

Thus, we defined the three non-overlapped time segments, being the *crisis* period when there is a sudden and persistent drop of $v_{i}\left(t\right)$ and a increase of volatility $\sigma_{i}\left(t\right)$. The dotted lines define starting and ending dates of crisis for SC and CO. In the first case, the start and end dates are 15 May 2008 to 15 March 2009 respectively, and for the second, are 13 February 2020 to 3 April 2020 respectively. For the pre-crisis and post-crisis periods, we take a time span equivalent to the duration of the crisis. For more details on the empirical analysis of volatility in these three time segments see Appendix A.

### 2.1. Spins’s Preservation of Returns Statistics

To model the state space of the system in different time periods, it is necessary to describe the probability distribution of this space from empirical data. Applying an energy based model using pairwise interactions [26] requires mapping the asset returns to a binary representation with values $s_{i}=+1$ and $s_{i}=-1$ indicating the state of asset *i*. We call each asset’s simplified representation a “spin,” similar to a magnetic dipole that can only have one positive or negative polarization. Thus, when the return of an asset $r_{i}\geq 0$, then the spin has value $s_{i}=+1$, and otherwise, $s_{i}=-1$ (or simply $s_{i}=\mathrm{sgn}\left(r_{i}\right)|r_{i}|$). Thus, since we will be working with a simplified version of the returns information, we are interested in verifying that the averages and correlations between each pair of assets are not lost by using this binary representation.

Let’s start by defining the average returns and spins for a time window *j* of size *T*, as:(1)$$\langle r_{i}\left(T_{j}\right)\rangle =\frac{1}{T}\sum_{t\in j}r_{i}\left(t\right)\;\mathrm{and}\;\langle s_{i}\left(T_{j}\right)\rangle =\frac{1}{T}\sum_{t\in j}s_{i}\left(t\right)$$

To simplify the notation, we will say that $\langle r_{i}\left(T_{j}\right)\rangle =\langle r_{i}\rangle$ and the same for spins $s_{i}$. The averages of returns and spins for each time segment are $\langle r_{T_{j}}\rangle$ and $\langle s_{T_{j}}\rangle$ for returns and spins respectively, so for the returns $\langle r_{T_{j}}\rangle =1/N\sum_{1}^{N}\langle r_{i}\rangle$ and for mean spin orientations $\langle s_{T_{j}}\rangle =1/N\sum_{1}^{N}\langle s_{i}\rangle$

As indicated by [27], the choice of *T* can be made considering the trade-off between noise and smoothing of the correlation coefficients. One can use *T* such that Q=T/N≥1 [19]. In this study we use T=250 days for the Subprime Case analysis (Q=250/10=25), which is roughly equivalent to one year of trading, and T=130 h for the Pandemic Case analysis (Q=130/15=8.67), which is roughly equivalent to one week of trading.

Figure 2a shows $\langle r_{T_{j}}\rangle$ considering the 10 country market indices of Table 1, and the means of the binarization of the daily log-returns $\langle s_{T_{j}}\rangle$. We observe correspondence between the two statistics. Figure 2b shows the same idea considering the 15 stocks of the Dow Jones (DJIA). If we compute for each of the *i*th stocks/indices the correlation between the series of returns $\langle r_{i}\left(T_{j}\right)\rangle$ and the series of their respective spins $\langle s_{i}\left(T_{j}\right)\rangle$, we obtain a measure of the linear correspondence between these two statistics. The idea is that these two statistics should be similar. For the case of the country indices, we obtain a correlation of 0.921 ± 0.045, while for the DJIA stocks it is 0.577 ± 0.115. All correlations between $\langle r_{i}\left(T_{j}\right)\rangle$ and $\langle s_{i}\left(T_{j}\right)\rangle$ are positive and significant, suggesting that the historical values of the mean orientation of the spins preserve the historical values of the means of the returns. Given the nature of the high frequency of data in the Pandemic case (hourly returns), the correlation, in this case, turns out to be lower, presumably due to the influence of noise structures present in the return series [28].

We compute correlation matrices over time for asset returns and spins’s orientations to show that spin orientations do not destroy the existing correlations between asset returns. Then we compare the correlations of both matrices, indicating a linear relationship between both statistics. We use normalized returns over a windows of size *T* periods which allows for a volatility-adjusted comparison of returns in each roll-window time [27]:(2)$$z_{i}\left(t\right)=\frac{r_{i}\left(t\right)-\langle r_{i}\left(T_{j}\right)\rangle }{\sigma_{i}\left(T_{j}\right)}$$
where $\sigma_{i}\left(T_{j}\right)$ is the standard deviation or volatility of returns $r_{i}\left(t\right)$ on the roll-window *j* of size *T*. The matrix Z of standarized returns with dimension N×T, is useful to define the asset correlation matrix:(3)$$R^{r}=\frac{1}{T}ZZ^{T}$$
with elements of $R^{r}$, $r_{ik}$∈[−1,1]. Similarly, we define the correlation matrix for the normalized spins $R^{s}$ using S, the matrix of spins with dimension N×T. To study whether the binarization of returns preserves the structure of linear correlations, we create a sequence of correlation matrices between returns $R^{r}$ and between spins $R^{s}$ in roll-windows of size *T*. We then compute the linear correlation between the elements of these two matrices. Again, the idea is that if the spins preserve the structure of correlations of returns, then this correlation should be positive and high. We call this measure $\rho_{T_{j}}$.

Figure 2c shows $\langle \rho_{T_{j}}\rangle$ between normalized returns and normalized spins for market indices in Subprime case (The behavior of the return correlations by region (Asia, Europe and North-America) is quite similar (not shown in the graph), although with different magnitude). Figure 2d shows the same for the indexes in Pandemic case. All correlations are positive over time and above 0.9 in the daily index data and above 0.7 in the hourly stock data.

The degree of co-movement order of the stocks can also be estimated through the mean-field entropy $S_{MF}$ [2,29]. This is an approach based on mean-field theory in which the complex problem of multiple components interacting with each other is reduced to taking into account the average effect of the other components on an individual. In this case we assume that the system is in equilibrium. Thus,
(4)$$S_{MF}\left(T_{j}\right)=-\sum_{i=1}^{N}\frac{1+\langle s_{i}\left(T_{j}\right)\rangle }{2}\ln\left(\frac{1+\langle s_{i}\left(T_{j}\right)\rangle }{2}\right)+\frac{1-\langle s_{i}\left(T_{j}\right)\rangle }{2}\ln\left(\frac{1-\langle s_{i}\left(T_{j}\right)\rangle }{2}\right)$$
which is computed for each roll window $T_{j}$, and $\langle s_{i}\left(T_{j}\right)\rangle$ is the average orientation of spin *i* in $T_{j}$. We take roll windows of 250 days and 130 h shifted by 1 day (or hour) for the Subprime and Pandemic case data respectively. Figure 2d,f reveals that in periods where the spins tend to be aligned in only one direction, i.e., average values diverging from zero (See Figure 2a,b), the entropy decreases. Conversely, when the average orientation of the spins is close to zero, i.e., maximum disorder, the entropy increases. These results agree with those of [2]. Low entropy levels are characteristic of times of crisis in which returns tend to move in a synchronized manner, while at high entropy levels, there is a lower degree of synchronization, evidenced by a decrease in correlations between assets. These results are descriptive and allow us to corroborate specific differences in orientations, pairwise correlations, and entropy in each of the three-time segments defined for each financial turmoil episodes.

### 2.2. MEP Model and Boltzmann Machine

In this section, we define the model to describe the state space of the system and the process to estimate the interaction parameters between spins.

#### 2.2.1. MEP and Ising Model

It is known that the maximum entropy distribution that is consistent with the first and second moments of the distribution of observable states can be described by:(5)$$p\left(s\right)=Z^{-1}\exp^{-\beta H(s)}$$
where $s=(s_{1},s_{2},.....,s_{N})$ is the representation of the state vector of each spin in an Ising system, Z is the partition function, H(s) is the energy or Hamiltonian of the system for an state s, and β is the inverse of the temperature of the system (for our analysis we will leave β=1). The vector s is the binary representation of the returns ($s_{i}=+1$ in case of a positive return, and $s_{i}=-1$ in case of a negative one), while the energy of the system H can be interpreted as the opposite of the utility function H(s)=−U(s) [2,30]. In equilibrium, the pairwise interaction model, which satisfies the MEP, gives rise to energy in the form of [10,31]:(6)$$H\left(s\right)=-\frac{1}{2}\sum_{i=1}^{N}\sum_{k=1}^{N}J_{ik}s_{i}s_{k}-\sum_{i=1}^{N}h_{i}s_{i}.$$
where the coupling $J_{ik}$ describes how spin *i* and *j* interact each other, and $h_{i}$ describes the tendency of spin *i* to be in a particular state. In Ising systems, this is called the magnetization and can be interpreted as the effect of external influences of the system on spin *i*. The set of all magnetizations for each spin is the vector h. The set of all possible couplings for the *N* spins is the coupling matrix J. Each of these measures describes the ferromagnetic ($J_{ik}>0$) or antiferromagnetic ($J_{ik}<0$) interaction between the pair of spins. In the former case, the spins tend to stay in the same state or move in the same direction, while in the latter case, the spins tend to have opposite states or move in the opposite direction. The elements of the diagonal $J_{ii}$ are null because they do not contribute to the energy of the system.

#### 2.2.2. Inference

The problem of finding coupling matrix parameters J and magnetizations h is called the inference or inverse Ising problem. A variety of methods are currently available for this (see e.g., [9,32]). We opt for a machine learning approach, using Boltzmann machines [33] which offers an accurate alternative to parameter estimation, but not necessarily fast in computation [34]. In this case, the Boltzmann machine must learn the parameters in successive approximations by minimizing the loss function. The loss function is the Kullback-Leibler divergence between the observed distribution and the one obtained from the model ($\mathrm{KL}(p_{obs}||p_{2})$). For this case, the Kullback-Leibler divergence will be the measure that indicates how different the distribution of the pairwise model is from the observed model.

We will consider the financial system with adaptive capacity in which it possesses the ability to change its parameters in order to remain in equilibrium in the face of changes in the environment [35,36]. Thus, it is possible to find these parameters assuming an equilibrium state, and they are able to describe the probability of states of the system p(s), such that it reproduces the observed statistics, mean orientation and pair-products:(7)$$\begin{array}{cc} {\langle s_{i}\rangle }_{model} & ={\langle s_{i}\rangle }_{obs}\\ {\langle s_{i}s_{k}\rangle }_{model} & ={\langle s_{i}s_{i}\rangle }_{obs} \end{array}$$
i.e., the mean orientation of spins and pair-products of the model are equivalent of the observed ones, for the *N* spins of the system. For this learning process, we follow the contrastive divergence process [33], in which the parameters are inferred through fits:(8)$$\begin{array}{cc} \delta h_{i} & =\eta \left({\langle s_{i}\rangle }_{obs}-{\langle s_{i}\rangle }_{model}\right)\\ \delta J_{ik} & =\eta \left({\langle s_{i}s_{k}\rangle }_{obs}-{\langle s_{i}s_{k}\rangle }_{model}\right) \end{array}$$
where η is the learning parameter. The statistics are calculated from the Metropolis-Hasting dynamics.

## 3. Results

In this section we describe three results. The first one is related to the inference process, in which we verify the level of consistency between the moments of the empirical and simulated spins, the analysis of the inferred parameters of the pairwise distribution, and its capacity to explain the system behavior. The second, concerning the level of frustration in the system, and the third, simulations to analyze the structure of the coupling network at the triad level. See Appendix B for the analysis of the correlations between assets and the stationarity of the return series, as a precondition before the inference process.

### 3.1. Couplings and Fields

#### 3.1.1. Consistency

The inference of couplings and fields is done for each of the three time slices previously defined as pre-crisis, crisis, and post-crisis in the case of Subprime and Pandemic. For each of the six time fragments, we used a Boltzmann machine with 25,000 steps to compute the approximations (Equation (Equation 8)), an initial learning rate of η=0.95 with a decay of 0.0004. In each of the machine steps, the estimators of mean orientations $\langle s_{i}\rangle$ and pairwise connections $\langle s_{i}s_{k}\rangle$ are computed via Metropolis-Hasting dynamics with 1500 steps. Consistency refers to ensuring that the machine successfully recovers the average orientation of the spins and the pairwise connection between assets. This is equivalent to testing Equation (Equation 7).

We can appreciate the consistency of the inference in Figure 3 that shows the comparison between the observed moments and those recovered from the simulations. It shows the relationship between the two-body connections $C_{ik}$ of the model and of the empirical observations. The Boltzmann machine manages to recover in an acceptable way the moments distribution of the spins in each time fragment. The Root Mean Squared (RMS) for all non-overlapped periods are in Table 2. All RMS’s range from a minimum of 0.012 to a maximum of 0.076. This suggests that the maximum entropy distribution (Equation (Equation 2)) inferred through the Boltzmann machine learning process is successful in recovering the orientations and covariances of the system states in moments of high and low volatility periods.

#### 3.1.2. Inferred Fields and Couplings

We study the inferred couplings for the three time segments for Subprime and Pandemic episodes with the Boltzmann Machine described previously. Figure 4 shows the results. The first thing to note is that we are in the presence of a system with negative and positive interactions. Second, it is worth analyzing the distribution of couplings. As we can see, the histogram recalls a certain resemblance to a Gaussian distribution. In the upper part of Figure 4 we illustrate a plot of the quantiles of the theoretical Gaussian distribution with that of the empirical distribution for each of the three non-overlapped time segments. As these quantiles lie on the diagonal, the distribution of the couplings conform to the Gaussian distribution. In both the Subprime and Pandemic cases, there is no excessive deviation of the inferred couplings from normality. However, some values in the lower tail and upper tail appear to deviate further from normality.

A more objective analysis suggests a Jarque-Bera normality test on the entire distribution of couplings for each of the three periods for the two cases. The results reveal that the normality hypothesis cannot be rejected. All *p*-values of each test are greater than 10%, except for the couplings in the pre-crisis period for Subprime case. In this episode, p=0.083. Then a $\chi^{2}$ normality test was done to ratify the above results. Again, all *p*-values of the test are greater than 10%, except in the post-crisis period for Subprime, which has p=0.014. Based on this evidence, it appears that the interaction strengths inferred in this period do not follow a normal distribution. However, both tests do not lead to reject the normality hypothesis by removing the coupling with lower intensity ($J_{ik}=-1.42$).

In all situations, the distribution of the interactions is asymmetric, i.e., $\langle J_{ik}\rangle >0$ indicating a predominantly ferromagnetic system. We note that, even when there are changes in the means of the coupling distribution, these variations are slight. For example in Subprime, the mean interactions decreased by 2.46% from pre-crisis to crisis, and then by 3.27% from crisis to post-crisis. However, in Pandemic, these numbers are an increase of 27.88% and a decrease of 14.38% respectively. During the crisis period in Pandemic, there was an increase in the intensity of interactions, and then a decrease in post-crisis. It is a similar result found by [9]; possibly a cause of herding behavior of the market and consequently the increase of correlations between financial assets. However, for the case of country indices we do not see an increase in the intensity of the couplings at the time of the Subprime crisis. The variations are very slight as previously indicated. In this sense, we want to emphasize that even with these slight variations in the level of ferromagnetism of the system, it is enough for the correlations in the crisis period to have an important increase with respect to pre-crisis and post-crisis (see Figure 2).

An interesting aspect is that the proportion of positive and negative interactions in the different periods remains virtually the same. For the Subprime, of the 45 couplings, 14 (31.1%) are negative, while in the Pandemic, of the 105 couplings, 80 (23.8%) are negative, except in the crisis period (here 79 are positive couplings and not 80). More interestingly, not all interactions keep their signs. For example in Subprime, from pre-crisis to crisis period, 15.6% of the couplings change signs from positive to negative, and the same proportion from negative to positive. From the crisis to the post-crisis period these proportions are exactly the same. In the case of Pandemic, from the pre-crisis to the crisis period, 19.4% of the couplings change sign from positive to negative, and 18.1% change sign from negative to positive. From crisis to post-crisis, 19.04% changed from positive to negative and 20.0% from negative to positive.

The above results suggest a certain similarity with the spin glass theory [37]. In this theory, the couplings are independent random variables chosen from a Gaussian distribution (with mean 0 and variance 1). Similar to what occurs in certain non-crystalline solids in which atomic bond structure is highly irregular, the magnetic state of the spins (magnetic orientation of the atoms) are characterized by being randomly aligned. It is assumed that the magnitudes of the interactions beetween spins are quenched and remain fixed for all time. In our case, we can consider this assumption valid at least during the observation period (quenched randomness [38]). Under this assumption we notice ferromagnetic and antiferromagnetic couplings between spins, and consequently, the presence of fustration [37,39]. This leads to multiple ground states and degeneracy for these low-lying states. We will deal with this evidence in Section 3.3.

Figure 5 reveals a positive relationship between interactions ($J_{ik}$) and correlations ($C_{ik}$). This relationship is weak, although significant given the value and the significance of the coefficients (b=0.295 in Subprime and b=0.356 in Pandemic periods, both significant at 0.1%). This positive relationship between pairwise connections and coupling has also been found in transactional datasets [34,40]. There does not seem to be a clear relationship between fields and the average orientation of the spins as suggested by Figure 5 (second column). It is worth noting that in the case of the Pandemic (Figure 5b) one can appreciate the difference in the correlations and in the average orientation of the spins in each of the three periods. In the crisis period (represented by the orange squares) the correlations are high and the average orientation of the spins is low with respect to pre and post-crisis. However, the same is not true for the couplings and fields. This can be interpreted as a manifestation that these quantities are inherent elements of the system and are responsible for the moments of the spins distribution (correlations and mean orientation of the spins).

#### 3.1.3. Pairwise Model Capacity’s Explanation

Since the MEP distribution is a model that considers pairwise interactions, we can study whether the pairwise correlations provide an effective description of the system in the different time segments with high and low volatility. Equation (Equation 5) describes the probability distribution of states of the maximum entropy system p(s), which is a function of the pairwise interactions. We will call this distribution $p_{2}\left(s\right)$. To evaluate the ability of this model to capture the intrinsic information in the system, we compute the Kullback-Leiber ($D_{\mathrm{KL}}$) divergence, between the empirical or observed distribution of states $p_{obs}\left(s\right)$, and the maximum entropy distribution, i.e., $D_{\mathrm{KL}}(p_{obs}∥p_{2})$. If, for example, $D_{\mathrm{KL}(p∥q)}=0$, then both distributions, *p* and *q* are equivalent in terms of information, i.e., there is no gain in using the distribution *p* as a description of *q*. On the other hand, we define $p_{1}\left(s\right)$ as the the distribution of a model that assumes that the spins are independent of each other, i.e., there are no pairwise interactions. In this model, only the $h_{i}$ fields are involved, so that $p_{1}\propto \exp\left(\sum_{i}h_{i}^{ind}s_{i}\right)$, where $h_{i}^{ind}={\tan h}^{-1}\left(\langle s_{i}\rangle \right)$ [41].

The Kullback-Leibler discrepancy between the observed model and the maximum entropy model ($D_{\mathrm{KL}}(p_{obs}∥p_{2})$) is equivalent to the difference in entropy between a model in which the spins of the system interact in pairs and a model in which the spins are driven independent of each other. This difference is also called multi-information gain $I_{N}=S\left(p_{1}\right)-S\left(p_{obs}\right)$. On the other hand, the entropy difference $I_{2}=S\left(p_{1}\right)-S\left(p_{2}\right)$ represents the amount of information due to second-order interactions. Thus, $I_{N}$ measures the amount of total correlation in the system due to higher order interactions greater than 2, and $I_{2}$ measures the contribution of 2nd order interactions in the system.

Let us define the quantity $G=I_{2}/I_{N}$. *G*-index is the fraction of entropy difference between the independent model and the data that are explained by the pairwise model [42]. If this quantity is close to 1 it indicates that the maximum entropy $p_{2}$ explains 100% of the system information through pairwise interactions. It should be clarified that the *G*-index is not the same as consistency (Section 3.1.1). The former is a measure of the ability of the Ising model to capture pairwise interaction information in the system. Consistency is the ability of the model to recover those pairwise correlations and mean spin’s orientations.

Figure 6 shows the densities of the Kullback-Leibler divergences in different periods for the Subprime and Pandemic periods. Each distribution was found based on 250 simulations with Metropolis-Hasting dynamics to find the distribution $p_{2}$ and $p_{1}$ using the inferred couplings and fields and then compared to the observed distribution of spins $p_{obs}$. In each situation, $D_{\mathrm{KL}}(p_{obs}∥p_{2})<D_{\mathrm{KL}}(p_{obs}∥p_{1})$, indicating that the pairwise distribution turns out to be a better representation of the observed information than that of an independent model, i.e., we gain in including the pairwise interactions to represent the behavior of the system.

Interestingly, the divergences for the Subprime and Pandemic periods undergo slight variations, indicating that in relative terms, the amount of information due to interactions remains relatively stable. For the Subprime case, we observe *G*-indices over 83%, which is indicative that pairwise distributions explain a good part of the available information, independent of the period. Additionally, changes in this measure between periods are virtually nonexistent. The increase in $D_{\mathrm{KL}}(p_{obs}∥p_{2})$ from pre-crisis to crisis is virtually nonexistent, and a slight decrease in $D_{\mathrm{KL}}(p_{obs}∥p_{1})$. From crisis to post-crisis the divergence in $p_{2}$ increases and so does the divergence with respect to $p_{1}$.

For Pandemic, the story is different; the pairwise distribution explains more than 50% of the available information, but it does not reach values as high as in Subprime. Additionally, in Pandemic, we notice an increase of $D_{\mathrm{KL}}(p_{obs}∥p_{1})$ and a slight decrease of $D_{\mathrm{KL}}(p_{obs}∥p_{2})$ from pre-crisis to crisis, indicating that in crisis, pairwise interactions seem to be better explaining the orientation of the spins than just assuming independent behavior from each other. Then, from crisis to post-crisis, the exact opposite occurs; slightly decreasing $D_{\mathrm{KL}}(p_{obs}∥p_{1})$ and increasing its counterpart $D_{\mathrm{KL}}(p_{obs}∥p_{2})$. We also see an increase in *G*-index during the crisis period, scraping 70%, and then decreasing post-crisis to values close to 60%. It seems to be in this case that in full crisis, the pairwise interactions capture more information than they do in *normal periods*.

As an additional comparison analysis, we performed an inference process with the Boltzmann machine throughout the lower volatility period to evaluate the KL divergences and G-index under these circumstances. For this, we used a new sample of daily prices from 22 June 2005 to 7 June 2006, for SC case and hourly prices from 11 November 2017, to 9 December 2017, for CO case (See Table 3 column “Low-Vol.”) For SC, the G-index continues to maintain an average value of 85%, but both divergences are slightly lower than other time segments. In contrast, for CO, we observe that the G-index is even lower, verging on 50% in this period, showing that the KL distance between the observed distribution and the pairwise MEP is even more significant than the other period under analysis.

### 3.2. External Influences

We can express the energetic contribution on each spin, identifying whether this contribution is a product of the interactions with each of the spins of the system or a product of the effective magnetic field. The energy function of the H system in the Equation (Equation 6) has two terms describing the energy contribution due to interactions between the spins and the energy due to fields or magnetizations. The latter is interpreted as the effect of external influences that tend to influence the orientation of the spins in one direction or another. It is expected that under conditions of high volatility, economic or health events will influence the behavior of the system giving preference towards a simultaneous alignment of the spins. In Figure 7a, we can see that the average orientation of the spins in the high volatility (crisis) time segment in both cases tends to be negative, which is presumably indicative of a common re-orientation of many components in one direction (in this case negative) triggered by external events.

As indicated by [8], the orientation of each index/stock $s_{i}$ is subject to an external bias $E_{i}^{\mathrm{ext}}=h_{i}\langle s_{i}\rangle$ and an internal bias $E_{i}^{\mathrm{int}}=0.5\sum_{k}J_{ik}\langle s_{k}\rangle$. The mean orientations $q_{i}=\langle s_{i}\rangle$ are taken for each time segment.

Figure 7b shows the relationship between the influence of external factors and interactions of other spins on each spin. Dots below the diagonal indicate that factors external to the system predominate over the spin orientation. Conversely, points above the diagonal indicate that the spin orientation is mostly influenced by the interaction of other spins in the system. We can observe that in the crisis period, most of the points are below the diagonal. In fact, in the crisis period, for 80% of the spins satisfied the condition ${\left|E\right|}_{i}^{\mathrm{ext}}>{\left|E\right|}_{i}^{\mathrm{int}}$, and 93.3% for the SC and CO case respectively. For SC and CO, the percentages are 40 and 33.3% in the pre-crisis periods and 30 and 46.7% in post-crisis respectively. A similar result is found in [9] in which the external field can become 10 times larger than the internal energy. These results confirm that the MEP pairwise model can capture the external influences on the behavior of the financial assets represented in our sample.

### 3.3. Frustration

Given the existing distribution of couplings, we can consider the system with the presence of disorder and frustration. In a disordered system there are positive and negative coupling values $J_{ik}$, as an independent random variables selected from a Gaussian distribution. We assume that the value of $J_{ik}$ are quenched, i.e., it is fixed in the analysis period, and the spins should be adjusted as they can according to those interactions. As indicated in Section 3.1.2, the system favors a ferromagnetic interactions with averages $\langle J_{ik}\rangle >0$ in each time period. The presence of positive and negative exchange interactions leads to competition and conflict in the system [38]. This configuration of couplings produces frustration because there will be states of the system that will not be able to satisfy all the bonds, thus, the system can be thought of as a spin glass.

Let us consider the coupling lattice G=(V,E), E=|E|×|E|, i.e., a complete graph, in which an edge e∈E connect a pair of vertices $\left(v_{i},v_{k}\right)$ with coupling $J_{ik}$. To study the frustration problem, we analyze subsets of the complete network of couplings by identifying triangles of *G*. In this case we define any triangle *T* as a subgraph of *G* possessing only three nodes $(v_{i},v_{j},v_{k})\in V$ through 3 edge couplings $\left(J_{ij},J_{ik},J_{jk}\right)$. There are four possible cases: two representing frustration, and two others without frustration. Figure 8a shows the situation.

In a complete graph, the number of triangles is N(N−1)(N−2)/6 so that for the case of Subprime crisis with 10 indexes the number of triangles is 120, and for the case of the Pandemic crisis, the DJIA with 15 stocks, the number of triangles is 455. Interestingly, the proportion of triangles in each class does not have too much variation in each of the three periods. Table 4 shows the number of triangles of each type in each case.

We observe that in both cases and all periods, the predominant triangle is the frustrated type D (two negative and one positive link). Likewise, the number of non-frustrated triangles type A and B are always very similar to the total number of type D triangles. Type C (3 negative links) do not exceed 4% of triangles in each period and in each crisis. In percentage terms, triangles represent, on average, 30, 18, 3 and 50 percent of the A, B, C and D types respectively for Subprime, and 42, 11.5, 1.3 and 45.5 percent for Pandemics. That is, independent of period and assets, we see a balanced proportion of frustrated and non-frustrated interactions. These results are interesting because they indicate that despite substantial increases in return volatility in the crisis period, the system maintains stable functional connectivities between assets. What seems to be subject to change are the intensities of these connectivities.

For each of the four classes of triangles, we computed the average of the intensity of the couplings. This is a simple way to describe the level of frustration of the system in different periods in the case of the Subprime crisis and the Pandemic. It is convenient to remember that as the average of these couplings becomes larger, we can interpret it as a tendency towards a higher level of ferromagnetism, while a decrease is equivalent to a tendency towards paramagnetism.

Figure 8b,c shows these averages for both cases (Subprime and Pandemic) in the three time segments. First, the average intensity of type A triangles is higher than that of type B, which is expected because in type A, all the couplings are positive, while in B, only one is positive. Second, in frustrated triangles, as expected, class D triangles have a higher average than class C triangles, because the latter has only one positive coupling and type D triangles have two. Third, the average of class triangles sorted from smallest to largest is C, B, D and A. One should not think that frustration leads to a decrease in the level of ferromagnetism. It is possible to observe that the system contains frustrated triangles with higher average intensity than unfrustrated triangles. It is sufficient to compare the averages of class D and B triangles. Fourth, the average intensities of the couplings in the three periods do not seem to have large variations. We observe that for type A triangles, there is a slight increase in the mean intensities from the pre-crisis to the crisis period, and then a slight decrease from the crisis to the post-crisis period. This pattern is also observed in type D triangles for DJIA assets in Pandemic. However this pattern of increase and then decrease does not seem to occur in the other types of triangles. For example, in types B and C, the average increases only occurs when moving from crisis to post-crisis with country indices, while it decreases with DJIA assets with type C. Thus, there does not seem to be a clear trend in the variation of intensities according to whether the triangles are frustrated or not.

### 3.4. Entropies of Triangle Structures

To gain further insight into the behavior of the system in periods of high and low volatility, we carried out simulations using the inferred parameters of the MEP distribution, which allows us to analyze the entropy of simpler structures of the coupling network. A deeper analysis of the triangle entropy reveals interesting aspects. For a network of *N* nodes, it is possible to define maximum entropy distributions that are consistent with correlations of order *K*, for K=1,2,...,N. For the case where K=N, we have the distribution that exactly describes the maximum complexity of interactions in the system. Currently we have estimated the maximum entropy distribution for K=2. In general, the entropies for each of these distributions, satisfies that $S_{N}\left(T\right)<S_{2}\left(T\right)<S_{1}\left(T\right)$ [43].

Now, let’s consider any triangle T∈G with nodes $(v_{i},v_{j},v_{k})\in V$. Each node is associated with a spin $s_{i}$, $s_{j}$ and $s_{k}$ respectively. The entropy of *T* is $S_{N}\left(T\right)=-\sum p\left(s_{i},s_{j},s_{k}\right)$$\log_{2}p\left(s_{i},s_{j},s_{k}\right)$. The entropy of *T* assuming independence is $S_{1}\left(T\right)=-\left(p\left(s_{i}\right)\log_{2}p\left(s_{i}\right)+p\left(s_{j}\right)\log_{2}p\left(s_{j}\right)+p\left(s_{k}\right)\log_{2}p\left(s_{k}\right)\right)$. On the other hand, we can find the entropy of any triangle *T* using the distribution describing the pairwise interactions (K=2). Using the pairwise model of the Equation (Equation 5), we can calculate the distribution for *T* as:(9)$$p_{2}\left(s_{i},s_{j},s_{k}\right)=\frac{1}{Z}\exp^{-J_{ij}s_{i}s_{j}-J_{ik}s_{i}s_{k}-J_{jk}s_{j}s_{k}-h_{i}s_{i}-h_{j}s_{j}-h_{k}s_{k}}$$
in this way, the pairwise entropy of the triangle is:(10)$$S_{2}\left(T\right)=\sum p_{2}\left(s_{i},s_{j},s_{k}\right)\log_{2}p\left(s_{i},s_{j},s_{k}\right)$$
This entropy is equivalent to that of the triangle *T* as if it were completely disconnected from the rest of the complete graph *G*. To calculate this entropy $S_{2}\left(T\right)$, we use the inferred $p_{2}$ distribution to simulate through a Metropolis-Hasting dynamics, a sample that reproduces the behavior of the spins in the domain of second-order interactions. Finally, the entropy $S_{N}\left(T\right)$ of a triangle *T* corresponds to the entropy of the observed joint distribution of the three spins i,j,k of the triangle.

Figure 9 shows the density of the entropies $S_{N}\left(T\right)$ of all 120 and 455 triangles existing in the complete network for the Subprime and Pandemic cases respectively in the three periods. In Subprime, the distribution of entropies are quite similar. In fact, the median for the pre-crisis, crisis and post-crisis periods are 1.854, 1.845 and 1.857 respectively. On the other hand, in the pandemic case, the situation is 1.980, 1.729 and 1.934, i.e., in the crisis period, the structures at the triangle level decrease their entropy in which there is evidence of higher level of synchronization in the orientation of the spins. Additionally, Figure 9 shows the relationship between entropies $S_{N}\left(T\right)$ and pairwise entropies $S_{2}\left(T\right)$ in each of the periods. All Pearson correlations between each pair of entropies is above 0.995 which indicates that the pairwise $p_{2}$ distribution correctly explains the network information by taking triangles as a structural basis with second order interactions.

Figure 10 shows the densities of $I_{2}\left(T\right)=S_{1}\left(T\right)-S_{2}\left(T\right)$ and $I_{N}\left(T\right)=S_{1}\left(T\right)-S_{N}\left(T\right)$ for Subprime Pandemic cases. The first difference $I_{2}\left(T\right)$ represents the contribution of pairwise interactions in the behavior of the spins, while the second difference $I_{N}\left(T\right)$ is the measure total correlation in the network. In Subprime, the densities of $I_{2}\left(T\right)$ and $I_{N}\left(T\right)$ are not very different between each of the three periods. However, there are small differences. In crisis, there is a slight increase in higher order interactions $I_{N}\left(T\right)$ with respect to pre and post-crisis, while pairwise interactions $I_{2}\left(T\right)$ decrease slightly. That is, in crisis, we observe a slight increase of K≥3 order interactions. In the Pandemic case, the densities are clearly distinguishable according to the period. In crisis, higher order interactions increase with respect to pre and post-crisis, while pairwise interactions do the same (The densities of $I_{N}\left(T\right)$ and $I_{2}\left(T\right)$ are more on the right side of the graph axis).

Figure 11 shows the measures of $I_{N}\left(T\right)$ and $I_{2}\left(T\right)$ according to the type of triangle. Recall from Figure 8 that triangles A and B are unfrustrated and type C and D are frustrated. It is interesting to note that the unfrustrated triangles possess higher levels of interactions of order K≥2 than the frustrated ones, although the former possess a higher degree of variance. This is particularly noticeable in the Subprime case.

Another interesting aspect is that in the Subprime case the medians of $I_{N}\left(T\right)$ are larger than those of $I_{2}\left(T\right)$, while in the pandemic case the opposite is true. That is, the structural components of the network at the triangle level tend to have more information in higher order correlations than pairwise correlations in Subprime, while in pandemic more information seems to be found at the pairwise correlation level. Note that these results are not in contradiction with the results of Table 3 because here we are comparing KL divergences between triangle types, and not between periods. In other words, under this analysis, we observe certain similarities between the two financial crises.

## 4. Discussion

This paper used the maximum entropy principle (MEP) to find a pairwise interaction model that describes the relationships between different sets of stock market assets in two distress episodes, namely Subprime crisis (SC), and COVID-19 outbreak (CO). We considered daily stock market price indices by country to investigate the SC case, and the hourly prices of firms belonging to the DJIA to investigate the CO case. This arrangement allows us to look at different assets at different times and with different data frequencies. We created three non-overlapped time segments for each crisis that we called pre-crisis, crisis, and post-crisis. Each of the segments represents periods that are characterized by low and high volatility. High volatility is precisely the time segment in which the crisis is triggered by a sudden and prolonged drop in asset prices.

Similar to what happens in a ferromagnetic material (particles oriented in one direction or another according to the influence of neighboring particles), the equity market’s overall behavior arises from the multiple interactions at the microscopic level between each asset. To find these stocks’ couplings, we carried out an inference process using Boltzmann machines to find a distribution model that replicates the first and second momentum of the stocks’ orientations or spins.

It is a stylized fact that the volatility of returns in periods of financial crisis has higher than normal levels [44]. This is captured by volatility measures for both cases in both the sets of assets. The variability of day-to-day returns in different time windows over crisis periods reveal dramatic increases compared to non-crisis periods, for both the Subprime and the Pandemic cases. It is worth noting evidence of agreement with the phenomenon of synchronization, in which the correlation of returns between assets tends to increase [45,46], and an entropy decreases [2]. This phenomenon of synchronization or co-movements of returns is relevant for stock markets since contagion generates a significant change in stock’s correlation coefficients. Economic structural similarities of countries and regions, coupled with global factors explain financial markets’ co-movement and generate financial contagion on a large scale [47]. Evidence indicates that interconnections among financial markets vary over time, being an uneven phenomenon among countries and regions [48]. Also, synchronization during financial turmoil periods is accompanied by rising implied volatility indices. As our results show, it is possible to gain a deeper understanding of the behavior of financial markets under turmoil episodes analyzing markets interactions that let to describe the relationships between different sets of financial assets. Accordingly, markets regulators and policy-makers should include these new perspectives and insights as input factors that conduct them to better supervise the proper functioning of financial markets, as well as to enhance the monitoring task of the financial system before new shocks again endanger the stability of global markets.

The pairwise interaction model is capable to recover the mean orientation of the spins and the pairwise connection between the assays in the crisis and non-crisis time segments for the Subprime and Pandemic cases. For example, when comparing the mean orientation of the empirical $\langle s_{i}\rangle$ spins to that of simulations made from the inferred distribution, the Root Mean Square errors (RMSe) are less than 0.0327, whereas the comparison between pairwise connections $C_{ik}$ of the empirical and simulated data the RMSe are less than 0.0355. These results are not dissimilar to those already found in studies using MEP in retailing and finance (see e.g., [2,8,9,34,40]). However, in this paper we focus on the extent the pairwise interactions explain the behavior of the system in periods of high and low volatility with different sets of assets.

The pairwise distribution of maximum entropy in the Subprime crisis using the daily market indices has a significant ability to explain the data. The second-order interactions explain more than 80% of the entropy in the system in each of the three-time segments. For the case of the Pandemic, the total amount of pairwise correlation is slightly higher than 50 percent, but not more than 70 percent, which still indicates that this type of interactions can explain a good part of the system information. In [2] these values can be as high as 98.5% for European country market indices, but using much more extended periods than we used here. They took 2253 hourly configurations over nine years and only six indices, while we considered segments of no more than 254 configurations and ten country market indices, and 10 stock indices. Of course, we are aware that by taking smaller sample sizes, we run the risk of not having enough samples to describe the system adequately ($M<<2^{N}$ being *M* sample size and *N* the number of assets), however, this is a recurrent problem in this type of studies even when taking larger sample sizes [4]. In spite of the latter, we find that the Boltzmann distribution inferred by MEP provides an acceptable representation of the system.

It is striking that the amount of pairwise correlation is higher at the country level than at the firm level. This may be explained by the fact that we used daily returns for the market indices, while for the DJIA firms we used hourly returns. Higher samplings frequency tends to produce lower correlations [49], while low samplings frequency tends to produce higher correlations. Thus, the amount of correlation captured by the pairwise distribution in the Dow Jones’s stocks is lower than that captured in the case of the country market indices. We conjecture that at the country level, structural innovations such as interest rates, inflation, fiscal policies and monetary policies with daily frequency take longer to be assimilated into the system. In contrast, other firm-level factors such as revenues, cost structures, investments, and innovations, which affect stock prices, are absorbed more quickly because they are incorporated into the stock prices measured at hourly frequency. Consequently, this could lead to higher-order interactions that the pairwise model does not capture. A natural extension of this work would be to investigate further the representation of financial system behavior with pairwise models at different measurement frequencies. This could provide further insight into the effect of measurement frequency on the replicability of state distribution moments based on MEP.

Overall, we find no evidence of a significant increase or decrease in pairwise information (as measured by the $D_{\mathrm{KL}}$ distance between the empirical distribution and the maximum entropy distribution) in none of the non-overlapped time segments, for both the Subprime and Pandemic cases. Likewise the variance and mean of the distribution of $D_{KL}(p_{obs}\left|\right|p_{2})$ remains stable in each of the time segments. In other words of amount of pairwise correlations does not seem to change significantly between periods of high variations of volatility of returns.

Another interesting finding is that the ratio of positive and negative couplings in each time segment and in both cases remains virtually constant even though several of them (approx. 15%) change sign. The systems during the Subprime and Pandemic cases are both predominantly ferromagnetic ($\langle J_{ik}\rangle >0$), and resemble spin glass with a normal distribution of couplings. This is not surprising considering that despite the diversity of countries and firms in the market, asset price changes depend on broadly the same macroeconomic signals [6]. This explains why correlations between assets are generally positive. However, we emphasize here that couplings are not the same as linear correlations. Correlations are produced by couplings, and in this sense, we observe that small changes in the magnitude of the couplings can produce large changes in the correlations between assets. In fact, in the crisis time segments, in each of the SC and CO cases, we observe a slight increase in the mean of the couplings. In other words, the system becomes more ferromagnetic, which explains at the physical level, why in times of financial turmoil, the returns between assets tend to move in tandem [9].

At the level of triangles or triads we observe that the structure of the network remains unchanged, being the frustrated structures type D the most abundant in both cases, followed by the non-frustrated triangles type A and B. It should be noted at this point in the process of counting the triangles, we have not discarded couplings, i.e., the network remains fully connected. However, by using significant cross-correlations, the number of frustrated triangles in the interaction network shows non-zero values just before and during times of financial crisis [50]. In fact, finding unbalanced triangles using cross-correlations is much less likely to occur in the network in periods of normality. This contrasts with our results, which suggest that the network of couplings based on the Ising $J_{ik}$ interactions possess a description of the system that as a whole, appears to remain less susceptible to structural change between periods of high and low volatility, unlike what occurs when studying the system with linear correlations.

## 5. Conclusions

The main contribution of this study is the analysis of the stock market using a pairwise model in two financial distress episodes of very different nature. Using a maximum entropy approach, it is possible to describe the amount of information due to second-order interactions for the Subprime crisis and COVID-19 outbreak, in different periods well distinguished by their level of volatility.

In summary, the main result of this study indicates that for subprime, the ability of the pairwise model to explain system information remains unchanged, regardless of whether it is a time of high or low volatility. This means that the amount of information due to second-order interactions remains virtually constant and explains a good portion of the interaction level in the market worldwide. On the other hand, when considering only the U.S. market in the COVID-19 outbreak, we found that in times of low volatility, the ability of the pairwise model to explain information is lower than in times of high volatility. On these periods, we observe a decrease in second-order interactions, indicating the presence of higher-order interactions. On the other hand, in high volatility periods, the second order interactions rises, which is associated with an increase in the correlations of returns between financial assets.

Relevant aspect of practical implications of our results, relates to the external influences to the system that governs the behavior of the spins. Consistent with the results of [9], we find that in high volatility periods, the average energy of the system due to external influences is higher than that of internal ones (due to interactions between spins). In contrast, during low volatility periods, the reverse is true. This means that in financial turmoil, the orientation of the spins is given primarily by agents external to the system. Consequently, at such times it is challenging to manage the financial risk of investment portfolios because of uncontrollable elements outside the financial system.

A comment should be made regarding the meaning of the $J_{ik}$ couplings. Technically, these parameters represent the Lagrange multipliers to the entropy maximization problem to find the distribution that best represents the moments of the system state distribution. But from the financial point of view, we understand that we are not dealing with a ferromagnetic material, but rather with a system involving hundreds of thousands of agents whose buying and selling decisions are reflected onto asset prices. In this sense we can think that the couplings are an integrated measure of inter-relationship between assets resulting from these thousands of human decisions, which are susceptible to biases and heuristics, escaping the full rationality assumptions of an economic agent.

It is well known that crisis periods are associated with an increase in the global synchronization of returns due to the collective dynamics of the economic system. Beyond this, we have been able to show that some additional characteristics of the financial systems can be understood in such periods using methods based on physical models. The Ising model is the simplest model to describe the collective dynamics of a complex system and is simple enough to describe the functional connections between each pair of system elements. Unlike most of the studies that are based on correlations and random matrix theory, the Boltzmann distribution, being that of maximum entropy, is flexible enough for studying financial systems from the entropy perspective.

Our work facilitates regulators, central banks, and policymakers, in the task of monitoring the synchronization of financial markets using entropy measures that complement the insights from other approaches such as correlation-based networks. As our results show, it is possible to gain a deeper understanding of the behavior of financial markets under turmoil episodes analyzing markets interactions that let to describe the relationships between different sets of financial assets. Accordingly, markets regulators and policy-makers should include these new perspectives and insights as input factors that conduct them to better supervise the proper functioning of financial markets, as well as to enhance the monitoring task of the financial system before new shocks again endanger the stability of global markets.

We think that major knowledge of financial systems is possible with this type of method. First, it is relevant to enhance the understanding of the equilibrium state of financial systems. In each inference process for each of the time segments under analysis, we assumed that the system is in equilibrium. This may not be fully valid. The question then arises as to what extent the financial system can be studied using the pairwise interaction distribution with couplings under circumstances where the system may be in a transition state. Second, it is worth analyzing the sensitivity of certain functional connectivities that could be determinant in the behavior of the overall dynamics of the financial system. Currently, we are not aware of studies that determine the impact of coupling changes on volatilities and correlations between financial assets.

## Figures and Tables

**Figure 1 entropy-23-01307-f001:**
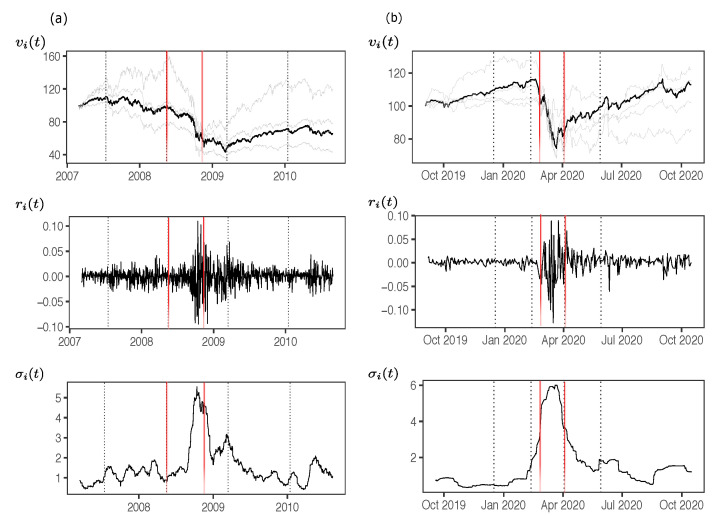
(**a**): Market indices for some countries in periods of interest. Reference value of 100 for all indices from 2 April 2007. Black dashed lines represent periods before crisis or pre-crisis starting from 15 July 2007 to until 14 May 2008, then crisis from 15 May 2008 to 15 March 2009, and post-crisis from 16 March 2009 to 15 January 2010. Red markers represent two events: On 9 August 2007, when BNP Paribas ceased its activities in its hedge funds of US mortgage market. This revealed that a large amount of derivative money was undervalued. Then on 15 September 2008, when Lehman Brothers declared bankruptcy. (**b**): Selected stocks belonging to the DJIA. Black dashed lines represent periods of pre-crisis starting from 17 December 2019 to 12 February 2020, then crisis from 13 February 2020 to 3 April 2020, and post-crisis from 4 April 2020 to 29 May 2020. There are two important events indicated as red markers: the first one on 29 February 2020 when Federal Reserve (Fed) Chairman Jerome Powell pledges to take steps to mitigate the economic impact of the virus outbreak. This comes in the midst of a steep drop in asset prices. Then in 9 April 2020 when the FED announces loans and economic aid to small and medium sized companies and local governments, to re-activate the economy. This marks the beginning of the recovery from the losses that occurred in the previous month.

**Figure 2 entropy-23-01307-f002:**
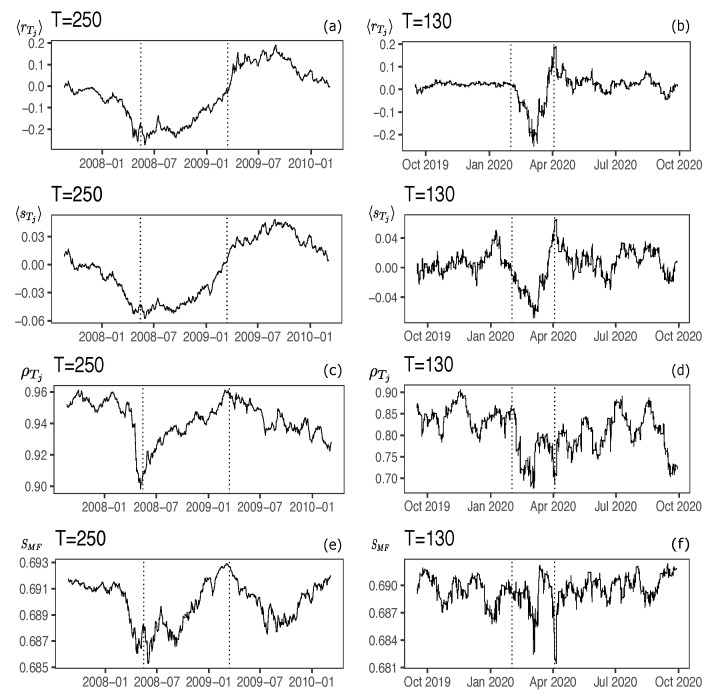
Means of returns $\langle r_{T_{j}}\rangle$ and spins $\langle s_{T_{j}}\rangle$ in roll-windows of size T=250 days for Subprime Crisis in (**a**) and T=130 h for COVID-19 outbreak in (**b**). Returns correlations $\rho_{T_{j}}$ between elements of the correlation matrix $R^{r}$ and of $R^{s}$ in roll-windows of size *T* for Subprime crisis in (**c**) and for COVID-19 outbreak in (**d**). The mean-field entropies $S_{MF}$ are shown in (**e**,**f**) for Subprime and Pandemic cases respectively. Notes: The vertical dotted lines indicates the period of crisis. For Subprime Crisis, the period of analysis is from 1 March 2007 to 30 August 2010. It includes the 10 country market indices. For COVID-19 outbreak, the dates are from 3 October 2019 to 9 October 2020. It includes the 15 stocks belonging to the DJIA indicated previously.

**Figure 3 entropy-23-01307-f003:**
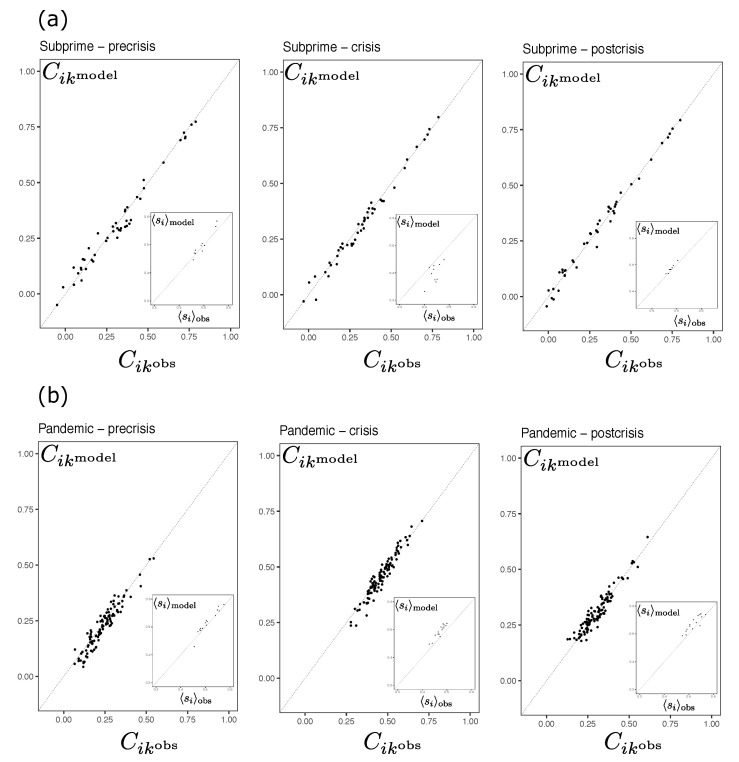
Scatterplots comparing the observed versus model estimations of two-body connections $C_{ik}=\langle s_{i}s_{k}\rangle -\langle s_{i}\rangle \langle s_{k}\rangle$ for each one of three non-overlapped periods (pre, crisis and postcrisis) for both, Subprime (**a**) and, Pandemic (**b**) cases. The grey dotted diagonal line represents the perfect correspondence between the two quantities. The insets of each figure compares the observed versus model estimations of mean orientations $\langle s_{i}\rangle$.

**Figure 4 entropy-23-01307-f004:**
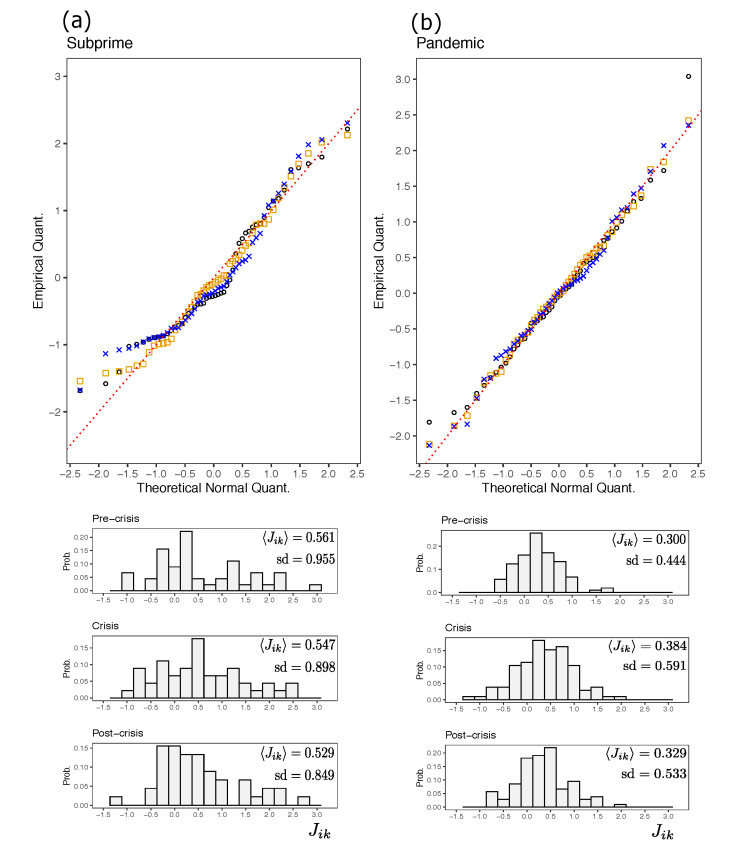
In (**a**,**b**) Comparison of the empirical coupling distribution for Subprime and Pandemic cases respectively, versus a theoretical Gaussian distribution. Black circles, orange squares and blue crosses represent quantiles of couplings in pre-crisis, crisis and post-crisis of the non-overlapping periods respectively. Histograms of the couplings are shown at the bottom of the plots. Means $\langle J_{ik}\rangle$ and the standard deviation sd are indicated.

**Figure 5 entropy-23-01307-f005:**
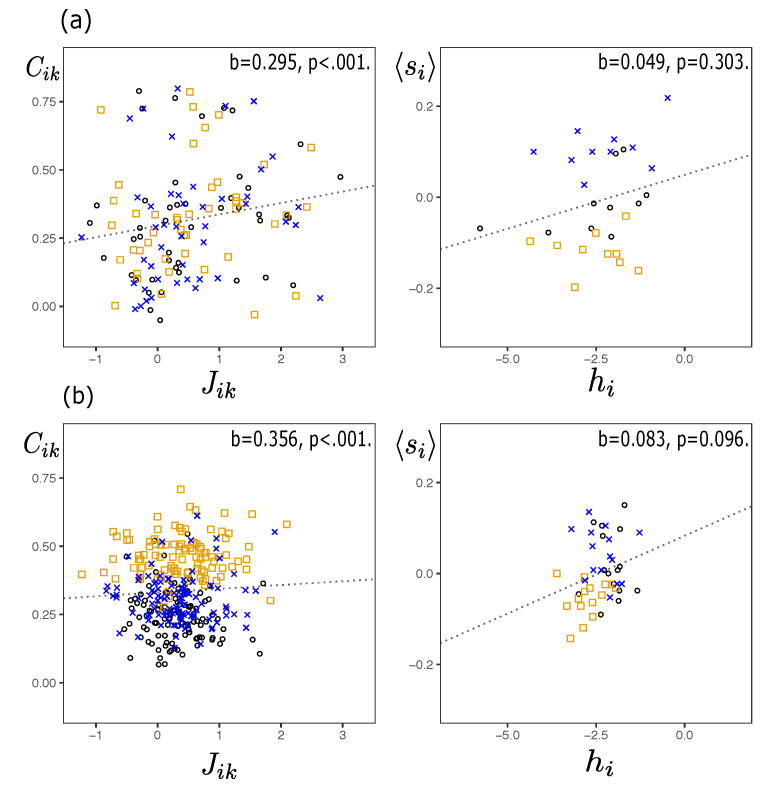
Comparison between inferred interactions $J_{ik}$ and two-body connections $C_{ik}$, between $h_{i}$ fields and mean orientations $\langle s_{i}\rangle$, for the Subprime (**a**) and Pandemic (**b**) cases. The black circles, orange squares and blue crosses represent, respectively, the pre-crisis, crisis and post-crisis non-overlapping periods. The dotted line represents the linear fit between the pair of variables.

**Figure 6 entropy-23-01307-f006:**
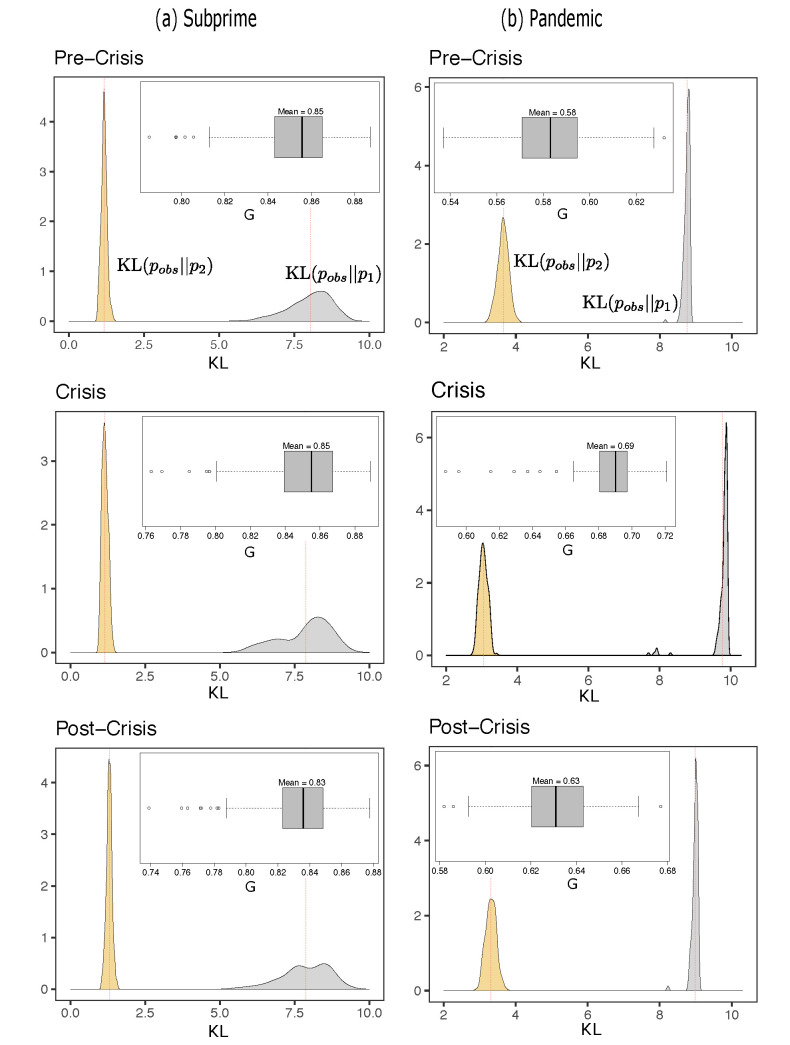
Densities of Kullback-Leibler (KL) Divergence between the observed and maximum entropy model ($D_{\mathrm{KL}}(p_{obs}∥p_{2})$) in light orange, and between the observed and the independent model ($D_{\mathrm{KL}}(p_{obs}∥p_{1})$) in gray. En column (**a**) and column (**b**) for the three periods of the Subprime and Pandemics cases respectively. Dotted red lines represents means for each distribution (also indicated in Table 3). The boxplot in the insets of each plot represents the distribution of *G*-index.

**Figure 7 entropy-23-01307-f007:**
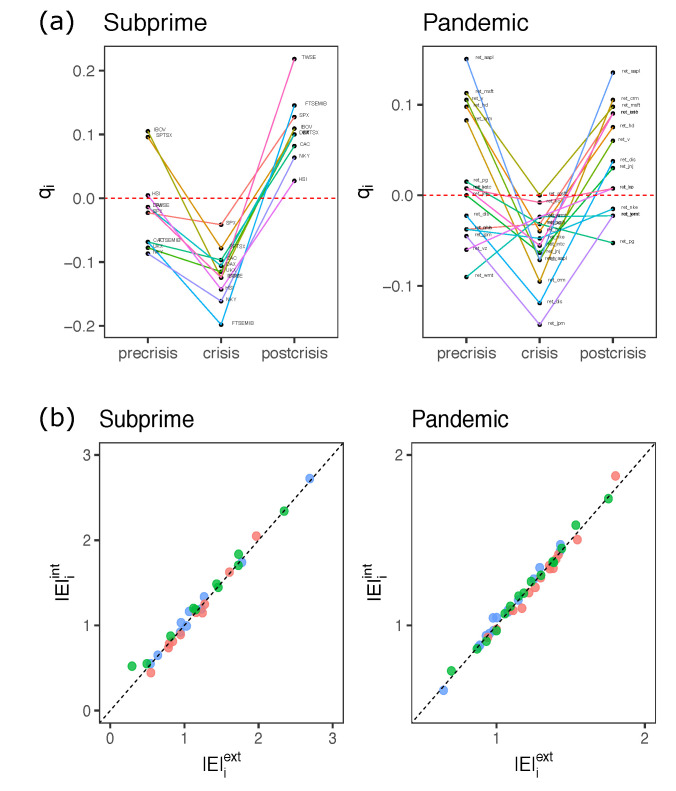
(**a**): Means of spin orientations $q_{i}=\langle s_{i}\rangle$ for each index/stock. (**b**): Scatterplot between external fields ${\left|E\right|}_{i}^{\mathrm{ext}}$ and internal energy ${\left|E\right|}_{i}^{\mathrm{int}}$ effects on each index/stock. Red, green and blue dots represent crisis, postcrisis and precrisis respectively. The diagonal represents equality between external field and internal energy.

**Figure 8 entropy-23-01307-f008:**
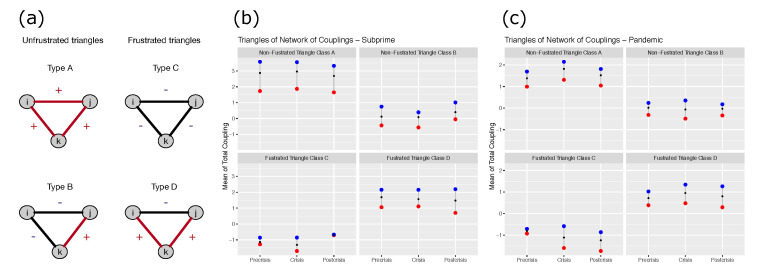
(**a**): Different possible classes of triangles on the complete network of couplings. Classes A and B are unfrustrated triangles, while classes C and D are frustrated ones. (**b**,**c**): Average of the coupling intensity for each type of triangle for the case of subprime crisis and COVID-19 outbreak respectively. Black dots represents the averages, blue and red ones, the maximum and minimum values respectively.

**Figure 9 entropy-23-01307-f009:**
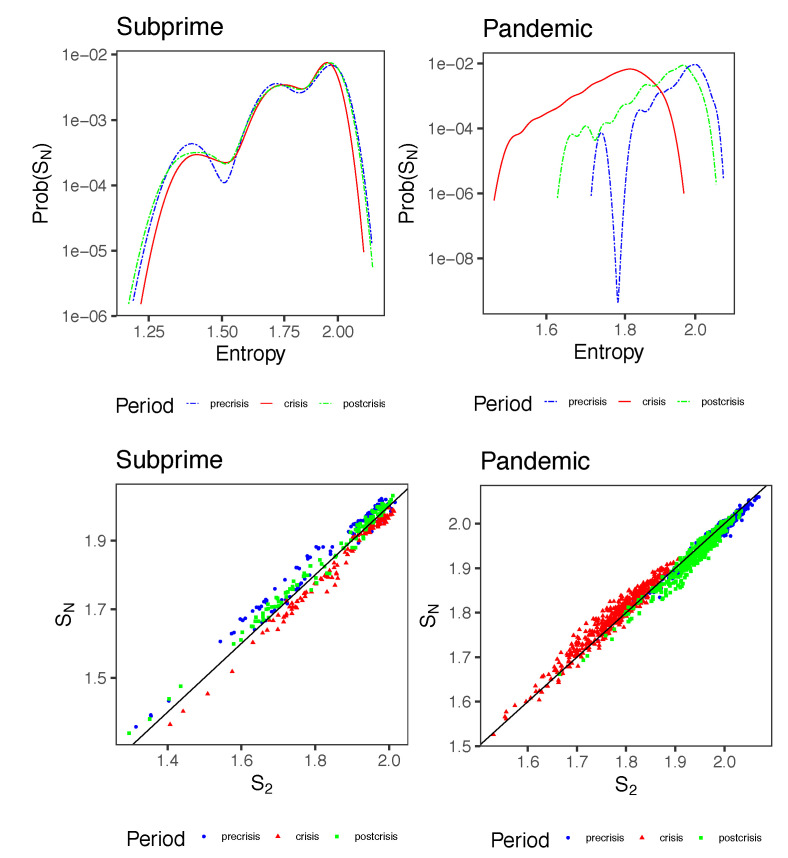
The first row shows the log-log density of $S_{N}\left(T\right)$ of triangles for the subprime and pandemic cases in each of the three periods. The second row shows scatterplots between $S_{N}\left(T\right)$ and $S_{2}\left(T\right)$ colored according to periods.

**Figure 10 entropy-23-01307-f010:**
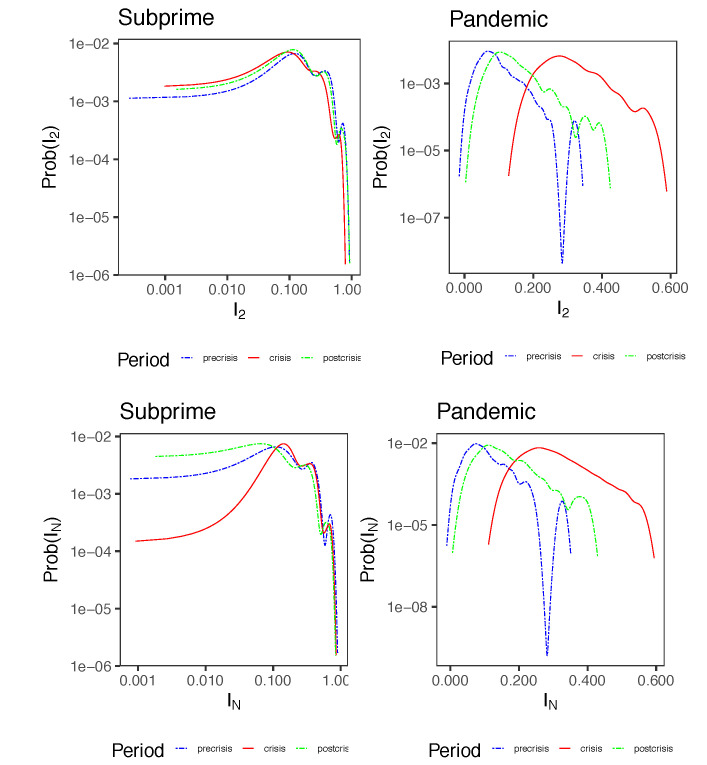
Densities of entropies differences $I_{2}\left(T\right)=S_{1}\left(T\right)-S_{2}\left(T\right)$ and $I_{N}\left(T\right)=S_{1}\left(T\right)-S_{N}\left(T\right)$ for Subprime and Pandemic cases for each of the periods.

**Figure 11 entropy-23-01307-f011:**
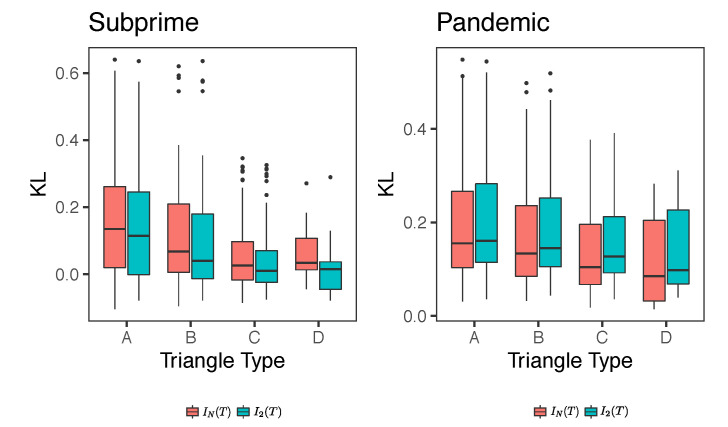
Boxplots of KL divergences $I_{N}\left(T\right)$ and $I_{2}\left(T\right)$ for each type of triangle for Subprime and Pandemic cases. Black bold line inside the boxplot indicates the median of KL measure.

**Table 1 entropy-23-01307-t001:** Country market indices and companies’ stocks analyzed for Subprime (SP) crisis (N=10 indices) and COVID-19 outbreak (CO) (N=15 stocks) cases.

Subprime Case	Pandemic Case
SPX (USA SP500)	MSFT (Microsoft Corp)
SPTSX (Canada)	AAPL (Apple Inc.)
IBOV (Brazil)	VZ (Verizon Communications Inc.)
UKX (UK)	INTC (Intel Corp)
CAC (France)	HD (Home Depot Inc.)
DAX (Germany)	CRM (Salesforce.com Inc.)
FTDEMIB (Milan)	WMT (Walmart Inc.)
NKI (Japan)	VIS (Visa Inc.)
HSI (Hong-Kong)	JPM (JPMorgan Chase & Co.)
TWSE (Taiwan)	JNJ (Johnson & Johnson)
	PG (Procter & Gamble Co.)
	NKE (NIKE Inc.)
	KO (Coca-Cola Co.)
	DIS (Walt Disney Co.)
	UNH (UnitedHealth Group Inc.)

**Table 2 entropy-23-01307-t002:** Root Mean Squared error (RMSe) for every defined non-overlapped period for Subprime and Pandemic cases, between the observed and recovered moments from the Pairwise model. Notations $\langle s_{i}\rangle$ indicates mean orientations, $\langle s_{i}s_{k}\rangle$ indicates pairwise connections and, $C_{ik}$ indicates two-body connections.

	Pre-Crisis	Crisis	Post-Crisis
Subprime:			
RMSe for $\langle s_{i}\rangle$	0.013	0.023	0.020
RMSe for $\langle s_{i}s_{k}\rangle$	0.076	0.044	0.018
RMSe for $C_{ik}$	0.032	0.025	0.024
Pandemic:			
RMSe for $\langle s_{i}\rangle$	0.032	0.026	0.027
RMSe for $\langle s_{i}s_{k}\rangle$	0.012	0.022	0.019
RMSe for $C_{ik}$	0.035	0.025	0.024

**Table 3 entropy-23-01307-t003:** Means of *G* index and Kullback-Leibler Divergence for each case and periods.

		Pre-Crisis	Crisis	Post-Crisis	Low-Vol.
Subprime	$D_{\mathrm{KL}}(p_{obs}∥p_{2})$	1.17	1.16	1.33	1.09
	$D_{\mathrm{KL}}(p_{obs}∥p_{1})$	8.03	7.88	8.87	7.21
	*G*	0.85	0.85	0.83	0.85
Pandemic	$D_{\mathrm{KL}}(p_{obs}∥p_{2})$	3.65	3.05	3.31	4.50
	$D_{\mathrm{KL}}(p_{obs}∥p_{1})$	8.75	9.77	8.98	8.80
	*G*	0.58	0.69	0.63	0.49

**Table 4 entropy-23-01307-t004:** Number of triangles of each type for different periods under analysis.

	Subprime			Pandemic		
Type	Pre-Crisis	Crisis	Post-Crisis	Pre-Crisis	Crisis	Post-Crisis
A	36	34	37	194	184	193
B	22	20	21	54	53	51
C	3	3	4	5	7	6
D	59	63	58	202	211	205

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
