# Peer review of "Equity Market Description under High and Low Volatility Regimes Using Maximum Entropy Pairwise Distribution"

_entropy, 2021, doi:10.3390/e23101307_

Round 1

Reviewer 1 Report

In this paper, the authors propose to analyse the relationship of comovements between assets, considering the Maximum Entropy Principle (MEP). Authors analyse two different crises (the subprime crisis and the financial outbreak due to the covid). The topic under analysis is relevant but I have some concerns:

  1. The topic needs a deeper literature review. Authors should decide if they want to make it independent of following the review which is started in the introduction. Although, the authors should compare their results with the findings of previous literature, making the needed comparisons.
  2. Authors should include other studies envolving the use of methodologies linked to entropy or other information theory measures, to analyse financial markets and, in particular, episodes of crises. See, for example, but not only, https://link.springer.com/article/10.1140/epjst/e2020-900124-y, https://doi.org/10.1016/j.jbankfin.2010.09.018.
  3. Finally, still in the literature review, authors should include works analysing the impact of covid in financial markets. A large strand of papers is available.
  4. My main concern is related with the fact that authors analyse two different groups of assets in the different crises. Why? During the subprime crisis authors analyse indices and in the covid outbreak individual assets. Why not always the same? I recommend the authors to replicate the analysis using both groups in both cases.
  5. Conclusions and discussion are relevant. I just recommend to add the information related with my previous concern.
  6. I found some language errors. Please, make a final review on the paper (just minor issues).

Author Response

Dear Reviewer:

We have addressed and responded to each of the comments and observations. Based on these observations, we have submitted a new version of the manuscript, which we hope is an improved version of the work, based on the reviewers' comments. In particular, we believe that the contextualization and connection with the financial system have been substantially improved.

Reviewer 2 Report

1. To study asset correlation matrices over time authors use rolling window normalized returns. Authors shuld clearly elaborate why they use this approach for obtaining corellation dynamics. There are many other approaches, such as multivariate GARCH models. This approach ins non-parametric and authors should argue why their approach is more appropriate for current study.

2. Term "asset interaction" in the abstract should be replaced with term "asset co-movements". It is more appropriate and clear.

3. Abstract should contain the main result of the study (not general conclusion)

4. When comparing pairwise dynamic correlation, in conclusion section, authors should, if possible, explain the differences in fluctuations between Europe, Asia, Southamerica and Northamerica. Accordingly, resuts should be more interpreted from real life point of view.

Author Response

(The authors gave the same response as above.)

Round 2

Reviewer 1 Report

I am satisfied with the changes made by the authors, even in the major concern I present. Despite not making my suggestion, it was added in possible future research, it is enough for me. Good job.